# Treatment of Smoldering Multiple Myeloma: Ready for Prime Time?

**DOI:** 10.3390/cancers12051223

**Published:** 2020-05-13

**Authors:** E. Bridget Kim, Andrew J. Yee, Noopur Raje

**Affiliations:** 1Department of Pharmacy, Massachusetts General Hospital, Boston, MA 02114, USA; ebkim@partners.org; 2Center for Multiple Myeloma, Massachusetts General Hospital Cancer Center, Boston, MA 02114, USA; ayee1@mgh.harvard.edu; 3Harvard Medical School, Boston, MA 02115, USA

**Keywords:** smoldering multiple myeloma, early intervention, prognostic features, risk stratification models

## Abstract

The current standard of care for smoldering multiple myeloma (SMM) is observation until there is end-organ involvement. With newer and more effective treatments available, a question that is increasingly asked is whether early intervention in patients with SMM will alter the natural history of their disease. Herein, we review the evolving definition of SMM and risk stratification models. We discuss evidence supporting early intervention for SMM—both as a preventative strategy to delay progression and as an intensive treatment strategy with a goal of potential cure. We highlight ongoing trials and focus on better defining who may require early intervention.

## 1. Introduction

First described in 1980, smoldering multiple myeloma (SMM) is an asymptomatic proliferative disorder of plasma cells that is a precursor to active symptomatic multiple myeloma [1,2]. SMM occupies the “gray” area between monoclonal gammopathy of undetermined significance (MGUS) and multiple myeloma (MM)—a malignancy of plasma cells traditionally defined by evidence of end-organ damage of hyper**c**alcemia, **r**enal dysfunction, **a**nemia, and/or **b**one lesions (also known as the “CRAB” criteria). Table 1 lists the diagnostic criteria of MGUS, SMM, and MM defined by the International Myeloma Working Group (IMWG) in 2014.

SMM represents a group of considerable heterogeneity. On one end of the continuum, there are patients whose disease behaves like MGUS with a low rate of progression to symptomatic MM. On the other end, patients develop evidence of end-organ damage with a risk of progression as high as 50–80% in the first 2 years of diagnosis [3,4,5,6,7]. The average risk of progression from SMM to MM is estimated to be 10% per year for the first 5 years [2]. In the following 5 years, this rate decreases to approximately 3% per year, and continues to decline over time to 1% per year for the next 10 years, at which point the progression rate is similar to MGUS. In all, the cumulative risk of progression is approximately 73% at 15 years [2]. However, what is lacking at this time is a molecular predictor from either genomic sequencing or gene expression profiling to assist in predicting which patients will progress to active MM [8]. Disease progression may be explained in part by other factors that are unrelated of myeloma cells, such as microenvironmental factors [9].

The current standard of care for SMM is close observation. In the past four decades, there have been significant efforts to refine the diagnosis of SMM [5]. Moreover, the increasing recognition of patient subsets with a high risk for progression and the availability of more effective and better tolerated drugs for MM have motivated investigations of early intervention to delay or arrest progression to symptomatic disease [5,10].

## 2. Evolving Definition of SMM

The first description of SMM included six patients who met the criteria for MM. This was based on the amount of serum monoclonal protein or degree of bone marrow involvement, and these patients did not have evidence of end-organ damage. They continued to remain stable without treatment, even after five years [1]. The authors concluded that treatment for SMM should be withheld as long as patients remain without symptoms. The risk of toxicities from the treatment at the time, melphalan and prednisone, appeared to outweigh any potential benefit [1]. In 2003, more than two decades since the initial description of SMM, the IMWG outlined diagnostic criteria of SMM (also described as “asymptomatic myeloma”) [11]. The definition of SMM included: Serum monoclonal protein of ≥3 g/dL and/or clonal bone marrow plasma cells of ≥10% with no end-organ damage in the form of hypercalcemia, renal dysfunction, anemia, or bone lesions. In addition, there was no evidence of amyloidosis, hyperviscosity, recurrent infections, or severe osteopenia. The recommendation at the time did not change and was to hold treatment until progression occurs.

In 2014, the IMWG revised the definition of MM and added a category of myeloma-defining biomarkers: ≥60% clonal bone marrow plasma cells; ratio of involved/uninvolved serum free light chain (sFLC) ≥100; or >1 focal lesion on magnetic resonance imaging (MRI) [12]. With the revised criteria, a subset of patients previously characterized as SMM but with a 2-year risk of progression of ≥80% now met the revised criteria for active disease requiring treatment [12]. This upstaged 10–15% of patients [5,12].

Cross-sectional imaging is an important modality in evaluating patients with plasma cell disorders. For a lytic lesion to be seen by a conventional skeletal survey using plain radiographs, over 50% of the bone needs to be involved [13]. Computed tomography (CT) is more sensitive than plain radiographs, and whole-body CT protocols using lower doses of radiation (given the high contrast of bone) have been evaluated in plasma cell disorders. In one study, low-dose whole-body CT (LDWBCT) detected lytic lesions in 25.5% of SMM and MM patients that were not present on a conventional skeletal survey [14]. Most of these missed lesions were in the pelvis, spine, ribs, and sacrum. In the recent consensus recommendations on imaging, the IMWG recommends LDWBCT over conventional skeletal surveys as the initial imaging method [15]. If the study is negative or inconclusive, further imaging is followed by whole-body MRI. Where this is not routinely available, MRI of the spine and pelvis or positron emission tomography-computed tomography (PET-CT) may also be used as an alternative to LDWBCT.

## 3. Prognostic Features and Risk Stratification Models for SMM

### 3.1. Risk Factors for Progression

Given the significant variability of SMM, a concerted effort has been made to identify prognostic features to better estimate the risk of progression. Many of these focus on biomarkers associated with increased tumor burden [5]. These factors include the type, amount, and rising levels of monoclonal protein; immunoparesis (suppression of one or more uninvolved immunoglobulins); abnormal sFLC ratio; and circulating plasma cells. Patients with IgA monoclonal protein SMM had a markedly shorter median time to progression (TTP) compared to those with other types of SMM. Median TTP of IgA-, IgG-, and light-chain only SMM were 27 months, 75 months, and 15.4 years, respectively [2,16]. Monoclonal protein level ≥4 g/dL was associated with median TTP of 18 months, whereas patients with lower monoclonal protein levels (<4 g/dL) had TTP of 75 months [2]. A rise in monoclonal protein levels over time also predicted the risk of progression. Patients whose monoclonal protein levels increased by 10% or greater on two consecutive evaluations (“evolving type”) had a TTP of 1.3 vs. 3.9 years in patients whose monoclonal protein levels remained stable [17]. Immunoparesis is a risk factor for progression. When two and one uninvolved serum immunoglobulin were suppressed, it corresponded to a TTP of 32 and 89 months, respectively, vs. 159 months in patients without immunoparesis [2]. An involved/uninvolved sFLC ratio was another risk factor for progression, with a median TTP of 30 months for an sFLC ratio ≥8 compared to 110 months for an sFLC ratio <8 [18]. Moreover, when the involved/uninvolved sFLC ratio was ≥100, TTP decreased to 15 months [19]. As mentioned earlier, an sFLC ratio ≥100 is a myeloma-defining event, and these patients are now reclassified as having MM [12]. Patients with a high level of circulating plasma cells, defined as >5 × 10^6^/L and/or >5% cytoplasmic Ig-positive plasma cells per 100 peripheral blood mononuclear cells, had a 2-year progression rate of 71% vs. 24% in patients with lower levels [20]. The extent of bone marrow involvement correlates with TTP in SMM [5]. Median TTP shortened with an increasing extent of bone marrow involvement with bone marrow plasma cells <20%, 20–50%, and >50% being associated with median TTP of 117, 26, and 21 months, respectively [2]. Bone marrow involvement ≥60% is now recognized as a myeloma-defining biomarker based on studies showing a 2-year risk of progression to approximately 90%, and these patients are now managed as active disease [12,19]. Additional risks to consider include an aberrant immunophenotype of bone marrow plasma cells as measured by multiparametric flow cytometry [21] and certain high-risk fluorescence in situ hybridization (FISH) features, such as del(17p), t(4:14), and t(14;16) [22]. Deletion 17p and t(4:14) are associated with the highest risk of progression, with a median time to progression of 24 months, as reported by the Mayo group [22]. Interestingly, hyperdiploidy, which is considered favorable in newly diagnosed multiple myeloma, has a higher risk of progression than a normal FISH pattern [22,23]. Subsequent studies suggest that some of these criteria, such as an sFLC ratio ≥100, may not be associated with as high a risk for progression as previously reported, emphasizing the challenges in using these risk factors [24,25].

A patient’s unique immune signature profile may play an important role for predicting progression. The prognostic value of immune profiling during minimal residual disease (MRD) assessment was investigated in the PETHEMA/GEM2010MAS65 study [26,27]. The investigators used the second-generation multi-parameter flow cytometry (MFC) assay to identify the distribution of 13 immune cell populations in the bone marrow of 146 elderly newly diagnosed MM patients following treatment. The immune cells quantified included myeloid progenitors, neutrophils, monocytes, eosinophils, mast cells, erythroid progenitors, erythroblasts, T-lymphocytes, natural killer T (NK-T) and NK cells, B-lymphocytes, and memory B-lymphocytes. Using this approach, three distinct immune signature clusters emerged: A (*n* = 16), B (*n* = 117), and C (*n* = 13). These clusters were separated by the steady rise in the numbers of erythroblasts and B-cell precursors, coupled with the steady decline in the numbers of naïve and memory B-cells [27]. Patients in cluster A showed a trend toward longer TTP as well as overall survival (OS). The OS values at 3 years were 100%, 65%, and 0% for clusters A, B, and C, respectively (*p* = 0.003) [27]. Their results show that the immune profiling during MRD assessment may be a relevant prognostic marker in identifying patients who might have prolonged disease control and survival even in the presence of MRD-positive disease. Using a similar platform in SMM may allow us to better risk stratify patients and identify who would be at a greater risk for progression.

Understanding the immune microenvironment may provide another key aspect of better determining progression risk. A study analyzing bone marrow immune cells in patients with MGUS and MM, as well as healthy donors, illustrated that there are progressive changes in the immune microenvironment landscape [28]. This included a progressive increase in terminal effector T cells with disease progression. The difference in the T cells in MGUS and MM was notable for the enrichment of stem-like memory T cells in MGUS, as opposed to T cells in MM, with greater expression of lytic genes and senescence markers. The loss of stem-like memory T cells in MM may in part explain the loss of immune surveillance when the disease becomes active.

The importance of integrating genomic analysis in predicting the risk of progression from SMM to active MM was highlighted in the SWOG S0120 study. Patients with MGUS or SMM (*n* = 331) were prospectively followed to assess the significance of clinical, genomic, and imaging prognostic features [29]. Data from the gene expression profiles (GEPs) of purified tumor cells showed that all molecular subtypes of active MM were also present in the asymptomatic precursor stage. Using a 70-gene signature, a GEP70 risk score of >−0.26 was correlated with an increased risk of progression. When the GEP70 risk score was combined with clinical prognostic features (elevated sFLC and M-protein), the progression was 67% at 2 years, potentially identifying a subset of patients with high risk for progression [29].

### 3.2. Risk Stratification Models

There are several models for estimating risk for progression in SMM. In the Spanish PETHEMA model, risk stratification is based on the abnormal/normal bone marrow plasma cells ratio and the presence of immunoparesis. Multiparametric flow cytometry is used to quantify aberrant bone marrow plasma cells defined as decreased CD38 expression, expression of CD56, and absence of CD19 or CD45 [21,30]. A predominance of these aberrant plasma cells (≥95%) correlated with a significantly higher risk for progression. High-risk patients (having both a predominance of aberrant bone marrow plasma cells and immunoparesis) had a 5-year rate of progression of 72%, while the 5-year progression rates for intermediate-risk (one risk factor) and low-risk (no risk factors) patients were 46% and 4%, respectively [21].

The Mayo Clinic 2008 model instead uses the amount of serum monoclonal protein (≥3 g/dL), degree of bone marrow involvement (≥10%), and sFLC ratio (involved/uninvolved ratio ≥8) to stratify SMM into three groups: High risk (all three risk factors), intermediate (two risk factors), and low risk (one risk factor) with associated 5-year progression risks of 76%, 51%, and 25%, respectively, and 2-year progression risks of 52%, 27%, and 12%, respectively [18]. The revised 2014 IMWG definition of SMM led to an update of the Mayo Clinic model. In the 2018 model, also known as “20/2/20,” the three risk factors are >20% involvement of bone marrow plasma cells, >2 g/dL serum monoclonal proteins, and sFLC ratio >20 [31]. The three risk groups are high risk (≥2 risk factors), intermediate risk (one risk factor), and low risk (no risk factor). The corresponding 2-year rates of progression for high, intermediate, low risk were 47.4%, 26.3%, and 9.7%, respectively [31]. Subsequently, this model was validated by IMWG in a separate cohort of more than 1000 patients and showed 2-year progression rates of 46%, 17%, and 5% in these groups [32]. When unfavorable cytogenetics identified by FISH, such as t(4;14), t(14;16), gain of 1q, or del(13q), were incorporated as risk factors, the 2-year rate of progression was increased to 59% in patients with ≥3 risk factors [32]. Table 2 summarizes the risk stratification models for SMM.

### 3.3. Limitations and Other Considerations

A challenge in risk stratification is the significant discordance between these models [33]. A prospective natural history study that included 77 SMM patients aimed to determine the degree of concordance between the two models (Mayo 2008 and PETHEMA) by comparing the distribution of patients with SMM classified as low, medium, and high risk for progression. The concordance between the two models was low, showing only 28.6% of patients belonging to the same risk group. Moreover, the difference was significant in classifying patients as low vs. high (*p* < 0.0001), low vs. non-low (*p* = 0.0007), and high vs. non-high (*p* < 0.0001) risk [33]. The low concordance rate highlights the challenges in interpreting and comparing clinical trials using different risk stratification models in SMM patients. Prospectively validated models may be needed to better characterize individual patient’s risk of progression to active MM [33].

Additionally, these risk stratification models assume a linear relationship between the disease burden and risk of progression [34]. However, whole genome analysis studies show that progression may not necessarily occur in a stepwise fashion. The mutational profiles and structural rearrangements were similar between paired samples of patients with SMM at the initial diagnosis and at the time of progression [35]. From this, two distinct progression patterns emerged: (1) A “static progression model”, in which the sub-clonal architecture and genomic features are preserved despite progressing to active disease, and (2) a “spontaneous evolution model”, in which a proliferative advantage is driven by additional change in the sub-clonal composition [35]. In support of the spontaneous evolution model are the findings from a National Cancer Institute (NCI) study analyzing serial serum samples of patients prior to progression to MM [36]. In this study, 37.2% of the patients progressed to high-risk MGUS from low/intermediate-risk MGUS in the span of 5 years, which suggests a progression model more dynamic than previously believed. In addition, a small proportion of patients directly progressed to MM from a low-risk MGUS, in line with a “spontaneous evolution model”.

## 4. Clinical Trials Investigating Early Intervention for SMM

Since SMM was initially described in 1980, there have been many attempts to answer if early intervention could delay disease progression. These included studies comparing observations vs. treatment with melphalan and prednisone [37,38], bisphosphonates [39,40], or thalidomide [41,42,43]. Compared to observation alone, these early studies, however, did not result in any noticeable clinical benefit. Interestingly, while treatment with pamidronate [40] or zoledronic acid [39] did not affect the time to progression, treatment with either bisphosphonate decreased the frequency of skeletal-related events at the time of progression.

The advent of more effective and better-tolerated drugs in MM has prompted readdressing early intervention in SMM. These strategies can be classified as a (1) prevention, which is designed as less intensive intervention aimed to delay progression; (2) treatment, with a goal of potential cure; and (3) other approaches that are aimed at directly engaging the immune system [5,30,44].

### 4.1. Prevention

The QuiRedex study by the Spanish Myeloma Group (PETHEMA) was the first large randomized study that evaluated the benefit of early intervention in SMM in the era of “novel” MM drugs. In this phase III trial, 119 patients who had high-risk SMM were randomized to either treatment with lenalidomide and low-dose dexamethasone (Rd) (*n* = 57) or observation alone (*n* = 62) [45]. High-risk SMM was defined as: (1) ≥10% plasma cells in the bone marrow and ≥3 g/dL serum monoclonal IgG protein; ≥2 g/dL serum monoclonal IgA protein; or ≥1 g urine Bence Jones protein over 24 h or (2) one of the two above criteria and ≥95% phenotypically aberrant plasma cells in the bone marrow by flow cytometry and presence of immunoparesis (suppression of >25% in one or more uninvolved immunoglobulins). Patients in the treatment arm received an induction regimen of lenalidomide 25 mg on days 1–21 with dexamethasone 20 mg on days 1–4 and 12–15 every 28 days for nine cycles, followed by a maintenance regimen of lenalidomide 10 mg on days 1–21 every 28 days for two years [45]. The primary endpoint was TTP to symptomatic disease. Following nine months of Rd induction, 79% of patients in the treatment arm achieved an overall response, which continued to increase to 90% in the maintenance phase. Moreover, the updated analysis showed that patients who received Rd had significantly longer TTP compared to observation alone; median TTP was not reached vs. 23 months, respectively (HR 0.24, 95% CI 0.14–0.41) [4]. Importantly, this study showed that early intervention led to improvement in the overall survival; 18% deaths was observed with Rd vs. 36% with observation with HR 0.43 (95% CI 0.21–0.92). The 3-year OS rates were 94% vs. 80% in the treatment and observation arms, respectively. At 5 years, OS was 88% vs. 71%, respectively.

The QuiRedex study was significant for demonstrating for the first time that early intervention improved TTP as well as OS in patients with high-risk SMM. However, the results of the study did not lead to a change in practice, due to some limitations in generalizing the study. First, it is possible that the control arm did not fare as well since it may have included patients who would now be reclassified as having active MM based on the updated 2014 criteria. Cross-sectional imaging with CT, MRI, or PET-CT, which is now standard practice, was not routinely performed at the time of the study, suggesting that patients with occult bone disease may have been included in the study. Second, in the observation arm, only 11% of patients who progressed received lenalidomide. This may have contributed to the differences seen in OS. Lastly, the specialized multiparametric flow cytometry used to define high-risk SMM is not readily available outside the trial centers, limiting the generalizability of the results [44].

Recently, the Eastern Cooperative Oncology Group revisited lenalidomide in SMM in a larger randomized study, E3A06. This study evaluated 182 patients with SMM and randomized patients to lenalidomide, as a single agent without dexamethasone (*n* = 90) vs. observation (*n* = 92) [44]. Eligible patients had SMM with bone marrow plasma cells ≥10% and an abnormal sFLC ratio. In the treatment arm, lenalidomide was given according to the conventional schedule, 25 mg daily for three weeks in a 4-week cycle. The primary endpoint was progression-free survival (PFS). To meet the requirement of disease progression, both biochemical disease progression and development of end-organ damage had to be present. Although the study began prior to the 2014 IMWG revised definition of SMM, most patients had no myeloma-defining events, thus meeting the current criteria. Unlike previous studies, the evaluation for bone involvement utilized more sensitive modern imaging, requiring MRI of the spine and pelvis at the time of study entry.

The overall response in the treatment arm was 50%, and as expected, no responses were reported in the observation arm. Early intervention with lenalidomide improved PFS vs. observation alone (HR 0.28; 95% CI 0.12–0.62). The PFS at one, two, and three years in patients receiving lenalidomide was 98%, 93%, and 91%. In the observation arm, the corresponding PFS was 89%, 76%, and 66%. The improved PFS was most pronounced in the high-risk category defined in the 2018 Mayo 20/2/20 model (*n* = 56; HR 0.09; 95% CI 0.02–0.44). Despite undergoing spine and pelvis MRI at screening, the most common cause of progression in the observation arm was bone-related progression (52.4%; 11 of 21 progressive disease cases). It would be important to know if these progressions were associated with symptoms, which is more clinically relevant, and how they were detected.

A total of six deaths were reported in the trial; two deaths occurred in the lenalidomide arm vs. four in the observation arm (HR for death 0.46; 95% CI 0.08–2.53). At this time, there is no significant OS difference observed between the arms. Thirty-six patients (41%) experienced grade 3–4 hematologic or nonhematologic adverse events in the lenalidomide arm, of which 25 patients (28%) had grade 3–4 nonhematological adverse events. The most common grade 3-4 adverse events were neutropenia (13.6%) and infection (10.2%). Details of the adverse events in the observation arm were not reported. The cumulative incidence of invasive second primary cancers at 3 years was 5.2% vs. 3.5% in the lenalidomide vs. observation arm, respectively. Eighteen (20%) of the lenalidomide-treated patients had treatment discontinuation and 80% of patients had a reduced dose. 

It is notable that nearly half (47.2%) of the patients had an MRI abnormality at study entry. It is unknown what proportion of these abnormal MRI findings would meet the definition of myeloma-defining events using the revised 2014 IMWG criteria, thus potentially upstaging these patients of having active MM. Similarly, baseline characteristics showed that some patients had myeloma-defining events under the revised definition of MM: 3.3% of the patients had ≥60% bone marrow plasma cells and 8.2% of the patients had an sFLC ratio >100. Since the greatest PFS benefit was observed with high-risk SMM, it would be of interest to know if the magnitude of the benefit decreases if these ultra-high-risk patients were excluded from the analysis (who are now considered active MM per the 2014 diagnosis criteria). Since patients with SMM do not have any symptoms (by definition), there is a greater premium placed on the tolerability of the intervention as well as demonstration of a benefit with the overall survival rather than only PFS. Despite a comparable quality of life between the groups, a significant portion of patients reported adverse drug events requiring therapy discontinuation or dose reduction. Table 3 summarizes the key trial characteristics of the QuiRedex and E3A06 clinical trials. 

### 4.2. Treatment and Other Ongoing Studies

On the other end of the spectrum is a more aggressive approach aimed at achieving a negative minimal residual disease (MRD) state with the hope of cure [5]. The hypothesis behind this approach is that SMM is where the best opportunity exists for a potential cure by inducing a deep response while the mutational burden is low and disease potentially more responsive to treatment [46]. Several early phase I/II trials investigating intensive regimens in SMM have reported their results. Table 4 shows the results of clinical trials investigating early intervention as a treatment strategy in SMM. In the phase II study investigating the use of carfilzomib, lenalidomide, and dexamethasone (KRd) in patients newly diagnosed with MM (NDMM) and high-risk SMM, patients received eight cycles of KRd induction, followed by lenalidomide maintenance for 2 years [47]. All 12 patients in the high-risk SMM cohort met the definition of high-risk disease based on the PETHEMA risk stratification model, and one patient also met the definition based on the Mayo risk stratification model. Skeletal survey and FDG-PET/CT were used to rule out osteolytic lesions. After a median follow-up of 15.9 months, 100% of the high-risk SMM patients achieved CR, 92% MRD negativity by multiparametric flow cytometry (95% CI 43–94%), and 75% MRD negativity by next-generation sequencing (95% CI 43–94%). Grade 3–4 adverse events were reported to be lymphopenia (50%), neutropenia (17%), rash (33%), infection (8%), and cardiac adverse events (8%). Of note, KRd in the NDMM cohort resulted in a lower CR rate of 56%, suggesting that the treatment may have induced a deeper response in high-risk SMM patients who may be more treatment sensitive.

In the phase II GEM-CESAR trial, six cycles of KRd induction were followed by high-dose melphalan and ASCT, and consolidation with KRd for two cycles, then Rd maintenance for 2 years in 90 patients with SMM whose risk was identified as >50% progression at 2 years [48]. With a median follow-up duration of 32 months, CR rates of 70% and 57% MRD negativity by next-generation flow were observed. PFS at 30 months was reported at 98%. Neutropenia (6%), thrombocytopenia (11%), infection (18%), and rash (9%) were among the grade 3–4 adverse events reported. Another trial evaluating treatment for active MM in SMM involved a combination of monoclonal antibody directed at SLAMF7, elotuzumab (Elo) with Rd [49]. In total, 50 eligible high-risk SMM patients received Elo-Rd induction for up to 8 cycles, followed by maintenance with Elo (monthly) and lenalidomide maintenance up to a total of 24 cycles. In 20 patients, high-risk cytogenetics features (presence of del(17p), t(4:14), and gain of 1q) were also detected by FISH. The results to date showed overall response rate (ORR) of 84% and PFS was not yet available. Most common adverse events reported were hypophosphatemia (34%), neutropenia (26%), and lymphopenia (22%).

An all oral regimen of lenalidomide, ixazomib (oral proteasome inhibitor), and dexamethasone (RId) may be a convenient option and was evaluated for early treatment benefit in SMM. In this phase II study, patients received RId induction for up to nine cycles, followed by lenalidomide-ixazomib maintenance up to 2 years [50]. Out of the 62 planned patients, 53 patients were enrolled, and 45 patients who completed at least one cycle were included in the analysis. High-risk cytogenetics features were present in 54% of the patients and 53.3% also met the definition of high-risk SMM based on the Mayo 2018 risk stratification model. With a median follow-up of 14.4 months, ORR was 91.1%, with CR 31.1% and VGPR 20% to date. Common grade 3 adverse events included hypertension (6.3%), hypophosphatemia (4.2%), and rash (4.2%), and grade 4 adverse events were thrombocytopenia (4.4%) and neutropenia (4.4%).

Daratumumab, an anti-CD-38 monoclonal antibody, is well tolerated and has shown single agent activity with deep and durable responses in both newly diagnosed and heavily pretreated MM patients. This formed the basis for the phase II CENTAURUS study that evaluated daratumumab monotherapy with varying treatment durations in SMM [51]. Patients (*N* = 123) with intermediate- or high-risk SMM were randomly assigned to one of the three daratumumab arms: Extended intense (progressively extended dosing from weekly, every 2, 4, and 8 weeks up to 20 cycles), extended intermediate (weekly in the first 8-week cycle, then every 8 weeks thereafter up to 20 cycles), and short dosing arms (one 8-week cycle of weekly dosing). After a median follow-up of 25.9 months, CR rates in the intense, intermediate, and short dosing arms were 4.9%, 9.8%, and 0%, respectively. The 2-year PFS rates in the same arms were 89.9%, 82.0%, and 75.3% (*p* < 0.0001). More grade 3–4 adverse events occurred in the intense arm vs. intermediate vs. short dosing arm (44%, 27%, and 15%, respectively). The most common grade 3–4 adverse events were hypertension and hyperglycemia. The ongoing phase III trial will further elucidate the benefit of daratumumab monotherapy in delaying progression in patients with SMM.

Some studies have examined treatment approaches that are aimed at directly engaging the immune system to prevent progression from SMM to active disease. These include a phase I study investigating PD-1 inhibition with pembrolizumab in patients with intermediate- and high-risk SMM [52]. Thirteen patients received pembrolizumab every 3 weeks for 8 cycles. Stringent CR was achieved in 8% and there were 85% SD and 8% PD. Three patients experienced immune-related adverse events and had discontinuation of therapy (2 transaminitis; one tubulointerstitial nephritis) [52]. It has been previously described that PD-L1 expression may increase the risk of progression from SMM to active disease [29,53]. Therefore, treatments directed at immune checkpoint inhibition and enhancing antitumor T-cell activities may play an important role in the immunogenic subgroup of patients. The role of engaging immune checkpoints in MM in combinations with immunomodulatory drugs like lenalidomide [54] or pomalidomide [55] is, however, on hold at the current time given unexpected worse outcomes but is being evaluated in other combinations. Another study aimed at engaging the immune response to prevent progression is a phase I/IIa study of a PVX-410 multipeptide vaccine with or without lenalidomide in 22 patients with intermediate- or high-risk SMM [56]. Twelve patients received 6 doses of PVX-410 vaccine alone (3 in low-dose, 9 in target-dose cohort) and 10 patients received 3 cycles of lenalidomide in addition to the vaccine. The immune response to PVX-410 was achieved in 95% (*N* = 20 evaluable patients; 10/11 patients (PVX-410 monotherapy); 9/9 patients (PVX-410 + R combination)). Most common adverse events were local injection site reactions and constitutional symptoms. These results provided a basis for further evaluation in a larger study. 

There are several ongoing clinical trials of intensive early intervention in SMM that have recently started accrual (Table 5). HO147SMM (NCT03673826) is a phase II study (planned *N* = 120) that randomizes patients with SMM to receive either a triplet regimen of carfilzomib, lenalidomide, and dexamethasone (KRd) or a doublet of lenalidomide and dexamethasone (Rd) for up to nine cycles. The patients will receive lenalidomide maintenance for up to 2 years. In the ASCENT trial (NCT03289299), the early intervention in SMM is intensified to a quadruplet regimen of daratumumab-KRd in the induction for up to 6 cycles, followed by consolidation with daratumumab-KRd for 6 more cycles, then daratumumab-lenalidomide maintenance for 12 cycles. The ECOG-ACRIN study (NCT03937635) randomizes patients with SMM (planned *N* = 288) into either daratumumab-Rd as a treatment approach or Rd as a prevention approach, with OS as the primary endpoint. Results from these and other trials are eagerly awaited and will better answer the question of which of these approaches is optimal in SMM. Other ongoing studies include a randomized trial comparing subcutaneous daratumumab vs. surveillance in the AQUILA study (NCT03301220) and the evaluation of isatuximab in high-risk SMM (NCT02960555). An ongoing study to illicit the immune system using the myeloma peptide vaccination approach with PVX-410 combined with an oral HDAC6 inhibitor citarinostat with or without lenalidomide (NCT02886065) is also underway.

## 5. Unanswered Questions Regarding Early Intervention for SMM

### 5.1. Who Requires Early Intervention?

In the modern era of novel drugs, more sensitive disease detection, and ongoing improvements in risk stratification, the question of who benefits the most from early intervention is still very relevant. As definitions and diagnostic criteria continue to evolve, our ability to accurately identify those with truly high-risk SMM with imminent progression to active disease will improve. Given the significant heterogeneous clinical course of SMM, current risk stratification models do not always reliably identify patients who will progress despite being classified as high risk. Moreover, a lack of concordance among risk stratification models is a challenge when interpreting and comparing studies for early intervention. There is a need to further our understanding of microenvironmental factors that contribute to disease progression, as well as refining other prognostic features that are not captured by current risk models. Until we can identify patients who progress to symptomatic MM with reasonable certainty or demonstrate an OS benefit, the balance of risk vs. benefit is unresolved.

### 5.2. What Is the Optimal Intensity of Early Intervention?

The E3A06 study of lenalidomide showed that the benefit in PFS was achieved despite not having a deep response from the treatment. Indeed, this was observed even with lenalidomide dose reductions and/or limited therapy exposure [44]. The authors of the study argue that achieving prolonged disease stability may not require a significant clonal burden reduction. This may be due to lenalidomide enhancing the immune function to promote long-term disease stability [44,57]. On the other hand, increasing the intensity of early intervention may offer a curative approach, rather than simply delaying progression. These questions need to be further explored in randomized clinical trials.

### 5.3. What Is the Optimal Duration of Early Intervention?

As with the intensity, the optimal duration of early intervention is yet to be elucidated. It is not known at this time how long pre-emptive therapy should be continued in the SMM stage, or if continuous therapy until progression is superior to a defined limited duration of therapy. In the E3A06 study, the authors suggested that based on the fixed duration of therapy in the Spanish QuiRedex and the median duration of therapy of 23 months in the E3A06 study, a duration of two years is the optimal length of therapy.

## 6. Suggested Approaches to the Management of SMM

The standard of care for patients with SMM today remains close observation, though this is an area subject to change in the future as ongoing studies report their findings. Despite some limitations, the current risk stratification models for SMM are important tools to estimate disease progression to active disease. Following the initial diagnosis of SMM, the IMWG recommends follow-up in 3 months. Stable patients can be followed up every 4 to 6 months for a year, and further extended to every 6 to 12 months [58]. Advanced imaging with LDWBCT, MRI, and/or PET-CT should be utilized to make a more accurate distinction between SMM and MM and to detect early progression during follow-up. Imaging should be performed annually [15]. After five years, the imaging could be stopped or the frequency reduced, especially in patients without high-risk features. Additional prognostic features, such as cytogenetics and flow cytometry information, may further refine individual risks for identifying patients with a high risk of progression. The management of patients with SMM with high-risk cytogenetic features, especially those harboring adverse abnormalities, such as del(17p) and t(4:14), is especially challenging. In the ECOG study, the numbers of patients with known high-risk FISH (defined as t(4:14) or deletion 17p) were too small, *n* = 7, to draw any conclusions. It may be prudent to continue close observation as a standard approach. The optimal endpoint for studies evaluating progression from SMM to active MM is OS or a substantial improvement in the quality of life. Many recent studies investigating early intervention in SMM incorporated high-risk cytogenetic features, as well as standard classification from risk stratification models. Results from these studies and mature data on OS from recent studies will provide further guidance in the broad incorporation of early intervention as standard practice in high-risk SMM in order to prevent or delay progression. Until more data are available, these patients should be followed very closely for evidence of progression. There is a need for a continued evaluation of the risks and benefits of early intervention for SMM, and these patients should be actively considered for enrollment in clinical trials.

## 7. Conclusions

Evidence supporting early intervention is emerging, but without any demonstration of an improvement in the overall survival, patients with high-risk SMM are best managed in the clinical trials setting until we have more data. Concurrently, there is a need to better understand the significant heterogeneity in SMM and to identify the patients who benefit the most from early treatment from those who can safely defer treatment.

## Figures and Tables

**Table 1 cancers-12-01223-t001:** Criteria for diagnosis of monoclonal gammopathy of undetermined significance, smoldering multiple myeloma, and multiple myeloma.

Monoclonal Gammopathy of Undetermined Significance	Smoldering Multiple Myeloma	Multiple Myeloma
Serum monoclonal protein <3 g/dL and Clonal bone marrow plasma cells <10% and Absence of end organ damage (CRAB criteria) or amyloidosis	Serum monoclonal protein (IgG or IgA) ≥3 g/dL or 24-hour urine monoclonal protein ≥500 mg and/or clonal bone marrow plasma cells 10–59% and No myeloma defining events (see below) or amyloidosis	Clonal bone marrow plasma cells ≥10% or biopsy proven plasmacytoma and Myeloma defining event: o End-organ damage (CRAB criteria) or o Biomarker of malignancy (one or more of the following) ▪ Clonal bone marrow plasma cell percentage ≥60% or ▪ Involved/uninvolved free chain ratio ≥100 (with involved free light chain ≥100 mg/L) or ▪ >1 focal lesion on MRI (≥5 mm)
Progression to multiple myeloma, solitary plasmacytoma, or AL amyloidosis: 1%/year	Progression to active multiple myeloma: 10%/year	
**End-organ damage (CRAB criteria) includes:** Hypercalcemia: calcium >1 mg/dL higher than the upper limit of normal or >11 mg/dL or Renal insufficiency: creatinine clearance <40 mL/min or creatinine >2 mg/dL or Anemia: hemoglobin >2 g/dL below the lower limit of normal or hemoglobin <10 g/dL or Bone lesions: one or more osteolytic lesions on skeletal radiography or CT		

**Table 2 cancers-12-01223-t002:** Summary of commonly the used smoldering multiple myeloma (SMM) risk stratification models.

Model	Risk Stratification		Progression Rate	Median Time to Progression
PETHEMA [21] ≥95% phenotypically aberrant plasma cells in bone marrow Immunoparesis	High risk	2 risk factors	5 year, 72%	23 months
	Intermediate risk	1 risk factor	5 year, 46%	73 months
	Low risk	No risk factors	5 year, 4%	Not reached
Mayo 2008 [18] Bone marrow plasma cells ≥10% Monoclonal protein ≥3 g/dL Free light chain ratio >8	High risk	3 risk factors	2 year; 52%; 5 year, 76%	1.9 years
	Intermediate risk	2 risk factors	2 year, 27%; 5 year, 51%	5.1 years
	Low risk	1 risk factor	2 year, 12%; 5 year, 25%	10 years
Mayo 2018 (20/2/20) [31] Bone marrow plasma cells >20% Monoclonal protein >2 g/dL Free light chain ratio >20	High risk	≥2 risk factors	2 year, 47.4%; 5 year, 81.5%	29.2 months
	Intermediate risk	1 risk factor	2 year, 26.3%; 5 year, 46.7%	67.8 months
	Low risk	No risk factors	2 year, 9.7%; 5 year, 22.5%	109.8 months

Free light chain ratio is defined as involved/uninvolved serum free light chain. Immunoparesis is a reduction below the lower limit of normal in the levels of one or two of the uninvolved immunoglobulins.

**Table 3 cancers-12-01223-t003:** Summary results of clinical trials investigating early intervention as a prevention strategy in SMM.

Clinical Trial	Phase	*N*	Intervention	Endpoints	Results	Adverse Events (Grade 3–4)
QuiRedex (NCT00480363) [4,45]	III	119	Rd (*n* = 57) vs. observation (*n* = 62) Induction: R 25 mg PO D1-21 + dex 20 mg PO D1-4, 12–15; every 28 days × 9 cycles Maintenance: R 10 mg PO D1-21; every 28 days × 2 years	Primary: TTP to active MM Secondary: RR, OS, safety	Median TTP (median follow up of 40 months): NR vs. 23 months; HR 0.24; 95% CI 0.14–0.41; *p* < 0.0001 OR 79% (induction phase); 90% (maintenance phase) 3-year PFS 77% vs. 30%; *p* < 0.001 3-, 5-year OS 94%, 88% vs. 80%, 71% HR 0.43; *p* = 0.03 Death 18% vs. 36%; HR 0.43; 95% CI 0.21–0.92	Infection (6%), asthenia (6%), neutropenia (5%), rash (3%)
E3A06 (NCT01169337) [44]	II/III	182	R (*n* = 92) vs. observation (*n* = 90) R 25 mg PO D1-21; every 28 days until progression	Primary: PFS Secondary: safety	OR 50% vs. 0%; 95% CI 39–61% 1-, 2-, 3- year PFS 98%, 93%, 91% vs. 89%, 76%, 66%; HR 0.28; 95% CI 0.12–0.62; *p* = 0.002 Death 6 vs. 2 patients; HR 0.46; 95% CI 0.08–2.53	Neutropenia (13.6%), infection (10.2%), skin rash (5.7%), dyspnea (5.7%), fatigue (6.8%), hypertension (9.1%), hypokalemia (3.4%)

CI = confidence interval; D = days; dex = dexamethasone; HR = hazard ratio; NR = not reached; OR = overall response; OS = overall survival; R = lenalidomide; Rd = lenalidomide/dexamethasone; TTP = time to progression.

**Table 4 cancers-12-01223-t004:** Summary of the results of clinical trials investigating early intervention as a treatment strategy in SMM.

Clinical Trial	Phase	*N*	Intervention	Endpoints	Results	Adverse Events (Grade 3–4)
NCT01572480 [47]	II	12	KRd induction → R maintenance Induction: K 20/36 mg/m^2^ IV D1, 2, 8, 9, 15, 16 + R 25 mg PO D1-21 + dex 20/10 mg (cycles 1–4/5–8) D1, 2, 8, 9, 15, 16, 22, 23; every 28 days × 8 cycles Maintenance: R 25 mg PO D1-21; every 28 days × 2 years	Primary: RR Secondary: MRD negativity, safety	100% CR (median follow up of 15.9 months 92% MRD-negative; 95% CI 62–100% (multiparametric flow cytometry) 75% MRD-negative; 95% CI 43–94% (next generation sequencing)	Lymphopenia (50%), neutropenia (17%), skin rash (33%), infection (8%), cardiac (8%)
GEM-CESAR (NCT02415413) [48]	II	90	KRd induction → HDT-ASCT → KRd consolidation → Rd maintenance Induction: K 20/36 mg/m^2^ IV D1, 2, 8, 9, 15, 16 + R 25 mg PO D1-21 + dex 40 mg D1, 8, 15, 22; every 28 days × 6 cycles HDT-ASCT: Melphalan 200 mg/m^2^ → ASCT Consolidation: K 36 mg/m^2^ IV D1, 2, 8, 9, 15, 16 + R 25 mg PO D1-21 + dex 40 mg D1, 8, 15, 22; every 28 days × 2 cycles Maintenance: R 10 mg PO D1-21 + dex 20 mg D1, 8, 15, 22; every 28 days × 2 years	Primary: MRD negativity Secondary: RR, safety	57% MRD-negative (next generation flow) 70% CR (median follow up of 32 months) 30-months PFS 98%	Neutropenia (6%), thrombocytopenia (11%), infection (18%), skin rash (9%)
NCT02279394 [49]	II	50	Elotuzumab-Rd induction → Elotuzumab-R maintenance Induction: Elotuzumab 10 mg/kg IV D1, 8, 15, 22 (cycles 1–2), D1, 15 (cycles 3–8) + R 25 mg PO D1-21 + dex 40 mg PO D1, 8, 15, 22 (cycles 1–2), D1, 8, 15 (cycles 3–8); every 28 days × 8 cycles Maintenance: Elotuzumab 20 mg/kg IV D1 + R 25 mg PO D1-21; every 28 days; cycles 9–24	Primary: PFS Secondary: RR, safety, TTP, OS	PFS—results pending ORR 84% (to date)	Hypophosphatemia (34%), neutropenia (26%), lymphopenia (22%)
NCT02916771 [50]	II	62 (planned)	RId induction → RI maintenance Induction: R 25 mg PO D1-21 + I 4 mg PO D1, 8, 15 + dex 40 mg PO D1, 8, 15, 22; every 28 days × 9 cycles (cycles 1–9) Maintenance: R 15 mg PO D1-21 + I 4 mg PO D1, 8, 15; every 28 days × 15 cycles (cycles 10–24)	Primary: PFS Secondary: TTP, ORR, OS, safety	45 patients who completed at least 1 cycle included in the analysis to date; median follow up of 14.4 months ORR 91.1%, CR 31.1%, VGPR 20% (to date)	Hypertension (6.3%), hypophosphatemia (4.2%), rash (4.2%), thrombocytopenia (4.4%), neutropenia (4.4%), hyperglycemia (2.2%)
CENTAURUS (NCT02316106) [51]	II	123	Daratumumab intense (*n* = 41) vs. intermediate (*n* = 41) vs. short dosing (*n* = 41) Daratumumab 16 mg/kg IV; every 8-week cycle (all schedule) Extended intense arm: Cycle 1: weekly Cycles 2–3: every 2 weeks Cycles 4–7: every 4 weeks Cycles 8–20: every 8 weeks Extended intermediate arm: Cycle 1: weekly Cycles 2–20: every 8 weeks Short dosing arm: Cycle 1: weekly × 8 doses	Primary: CR, PD/death rates Secondary: ORR, PFS, OS	Intense vs. intermediate vs. short dosing (median follow up of 25.9 months): CR rates: 4.9%, 9.8%, 0% PD/death rates: 0.059 (80% CI 0.025–0.092), 0.107 (80% CI 0.058–0.155), 0.150 (80% CI 0.089–0.211) 2-year PFS: 89.9% (90% CI 78.5–95.4%), 82.0% (90% CI 69.0–89.9%), 75.3% (90% CI 61.1–85.0%); *p* < 0.0001	Grades 3–4 TEAEs: 44%, 27%, 10% for intense, immediate, short dosing, respectively. The most common grade 3–4 TEAEs (observed in >1 patient in any arm): hypertension, hyperglycemia
NCT02603887 [52]	I	13	Pembrolizumab Pembrolizumab 200 mg IV every 3 weeks × 8 cycles; with option to continue up to 24 cycles if continued benefit	Primary: ORR Secondary: clinical benefit rate (≥minor response), safety	sCR 8%, SD 85%, PD 8%	Discontinuation due to irAEs: 3 patients (2 transaminitis; 1 tubulointerstitial nephritis)
NCT01718899 [56]	I/IIa	22	PVX-410 vaccine ± R Low-dose cohort (*n* = 3): PVX-410 vaccine 0.4 mg SC; every 2 weeks × 6 doses Target-dose cohort (*n* = 9): PVX-410 vaccine 0.8 mg SC; every 2 weeks × 6 doses PVX-410 + R combination cohort (*n* = 10): PVX-410 vaccine 0.8 mg SC; every 2 weeks × 6 doses + R 25 mg PO D1-21; every 28 days × 3 cycles	Primary: AEs Secondary: immune response to vaccine	Most common AEs—injection site reactions, constitutional symptoms Immune response to PVX-410 achieved in 95% (*N* = 20 evaluable patients; 10/11 patients (PVX-410 monotherapy); 9/9 patients (PVX-410 + R combination)	

AEs = adverse events; CI = confidence interval; CR = complete response; D = day; dex, d = dexamethasone; HDT-ASCT = high-dose therapy/autologous stem cell transplantation; I = ixazomib; irAEs = immune-related adverse events; K = carfilzomib; MRD = minimal residual disease; ORR = overall response rate; OS = overall survival; PD = progressive disease; PFS = progression free survival; R = lenalidomide; RR = response rate; sCR = stringent complete response; SD = stable disease; TEAEs = treatment-emergent adverse events; TTP = time to progression.

**Table 5 cancers-12-01223-t005:** Selected other ongoing clinical trials investigating early intervention in SMM.

Clinical Trial	Phase	*N*	Intervention	Endpoints	Results	Study Dates
HO147SMM (NCT03673826)	II	120 (planned)	KRd vs. Rd → R maintenance KRd arm: K 20/36 mg/m^2^ IV D1, 2, 8, 9, 15, 16 + R 25 mg PO D1-21 + dex 20 mg (cycles 1–4), 10 mg (cycles 5–9) D1, 8, 15, 22; every 28 days × 9 cycles Rd arm: R 25 mg PO D1-21 + dex 20 mg (cycles 1–4), 10 mg (cycles 5–9) D1, 8, 15, 22; every 28 days × 9 cycles Maintenance: R 10 mg PO D1-21; every 28 days × 2 years	Primary: PFS Secondary: MRD negativity; OS, safety	On-going	Study start date: Nov 19, 2018 Estimated primary completion date: Oct 2025 Estimated study completion date: Oct 2025
ASCENT (NCT03289299)	II	83 (planned)	Dara-KRd induction → Dara-KRd consolidation → Dara-R maintenance Induction: Dara 16 mg/kg IV D1, 8, 15, 22 (cycles 1–2); D1, 15 (cycles 3–6) + K 20/36 mg/m^2^ IV D1, 2, 8, 9, 15, 16 + R 25 mg PO D1-21 + dex 40 mg D1, 8, 15, 22; every 28 days × 6 cycles Consolidation: Dara 16 mg/kg IV D1 (cycles 7–12) + K 36 mg/m^2^ IV D1, 2, 8, 9, 15, 16 + R 25 mg PO D1-21 + dex 20 mg D1, 8, 15, 22; every 28 days × 6 cycles Maintenance: Dara 16 mg/kg IV D1 (every odd cycle for cycles 13–24) + R 10 mg PO D1-21; every 28 days × 12 cycles	Primary: sCR Secondary: MRD negativity; OS, PFS, safety	On-going	Study start date: May 25, 2018 Estimated primary completion date: Jun 1, 2022 Estimated study completion date: Jun 1, 2026
NCT03937635	III	288 (planned)	Dara-Rd (treatment approach) vs. Rd (prevention approach) Dara-Rd arm: Dara 16 mg/kg IV D1, 8, 15, 22 (cycles 1–2); D1, 15 (cycles 3–6); D1 (cycles 7–24) + R 25 mg PO D1-21 + dex 40 mg D1, 8, 15, 22 (cycles 1–12); every 28 days up to 24 cycles Rd arm: R 25 mg PO D1-21 + dex 40 mg D1, 8, 15, 22 (cycles 1–12); every 28 days up to 24 cycles	Primary: OS, FACT-G score Secondary: PFS, best response, safety	On-going	Study start date: Apr 30, 2019 Estimated primary completion date: Nov 30, 2023 Estimated study completion date: Nov 30, 2028
AQUILA (NCT03301220)	III	390 (planned)	Daratumumab SC vs. observation Daratumumab 1800 mg SC D1, 8, 15, 22 (cycles 1-2); D1, 15 (cycles 3–6); D1 (cycles 7–39) every 28 days up to 3 years	Primary: PFS Secondary: TTP, ORR, OS, safety	On-going	Study start date: Nov 7, 2017 Estimated primary completion date: Dec 16, 2021 Estimated study completion date: Dec 22, 2025
NCT02960555	II	61 (planned)	Isatuximab Isatuximab IV D1, 8, 15, 22 (cycle 1); D1, 15 (cycles 2–6); D1 (subsequent cycles); every 28 days up to 30 cycles	Primary: RR Secondary: PFS, OS	On-going	Study start date: Feb 8, 2017 Estimated primary completion date: Feb 1, 2022 Estimated study completion date: Feb 1, 2022
NCT02886065	Ib	20 (planned)	PVX-410 vaccine + citarinostat ± R PVX-410 + citarinostat + R arm: PVX-410 vaccine 0.8 mg SC biweekly × 6 doses + citarionostat 180 mg PO D1-21 × 3 cycles + R 25 mg PO D1-21 × 3 cycles; every 28 days PVX-410 + citarinostat arm: PVX-410 vaccine 0.8 mg SC biweekly × 6 doses + citarionostat 180 mg PO D1-21 × 3 cycles; every 28 days	Primary: safety	On-going	Study start date: Nov 2016 Estimated primary completion date: May 2021 Estimated study completion date: May 2021

ASCT = autologous stem cell transplantation; D = day; Dara=daratumumab; dex, d = dexamethasone; FACT-G = functional assessment of cancer therapy-general; K = carfilzomib; R = lenalidomide; MRD = minimal residual disease; ORR = overall response rate; OS = overall survival; PFS = progression free survival; RR = response rate; sCR = stringent complete response; TTP = time to progression.

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
