# Peer review of "Treatment of Smoldering Multiple Myeloma: Ready for Prime Time?"

_cancers, 2020, doi:10.3390/cancers12051223_

Round 1

Reviewer 1 Report

The manuscript by Kim et al., is a detailed and well written review on a very important topic in the field of multiple myeloma, concerning the question of treatment for smoldering multiple myeloma patients based on risk versus the current standard of close observation.

The manuscript is well organized and includes critically informative sections for risk stratification, ongoing clinical trials evaluating new treatment options for SMM, and remaining open questions for early treatment of SMM.

The two tables included are useful and informative. While a third table of ongoing trials could have been included, the section covering this topic is sufficiently informative such that this is not necessary.

The primary omission from the review article was an update on the results of some of the newer clinical trials in progress to bolster section 4.2. It would be informative and very useful to provide (1) a table of trials in progress, and (2) importantly, a summary - preferably in the text, or in the table, of any results to date from early Phase 1/2 trials. This would significant enhance the value of this manuscript.

No proposed major edits or inclusions. Very well prepared review article.

Author Response

Thank you for your thoughtful review and comments. Please see below response:

Reviewer 1:

The manuscript by Kim et al., is a detailed and well written review on a very important topic in the field of multiple myeloma, concerning the question of treatment for smoldering multiple myeloma patients based on risk versus the current standard of close observation.

The manuscript is well organized and includes critically informative sections for risk stratification, ongoing clinical trials evaluating new treatment options for SMM, and remaining open questions for early treatment of SMM.

The two tables included are useful and informative. While a third table of ongoing trials could have been included, the section covering this topic is sufficiently informative such that this is not necessary.

  • Addressed – Table 5 (on-going clinical trials investigating early intervention in SMM) has been added

The primary omission from the review article was an update on the results of some of the newer clinical trials in progress to bolster section 4.2. It would be informative and very useful to provide (1) a table of trials in progress, and (2) importantly, a summary - preferably in the text, or in the table, of any results to date from early Phase 1/2 trials. This would significant enhance the value of this manuscript.

  • Addressed- Table 3 (summary of SMM trials as prevention approach) & Table 4 (summary results of SMM trials as treatment approach) have been added, including updated results to date from phase 1/2 trials. Section 4.2 text has been expanded.

No proposed major edits or inclusions. Very well prepared review article.

Reviewer 2 Report

The authors reviewed and discussed the treatment strategy for smoldering multiple myeloma, especially focused on the novel agents in recent years. The manuscript is well prepared and have referenced updated data. However, there is still some issues to be solved before publication.

1.  Though the authors have reviewed and referenced updated studies, the manuscript is too short and not qualified as a full-review articles in this journal. The authors should expand the text for more information.

2. The authors aimed for the new treatment strategy in the era of novel agents, however, in addition to lenalidomide, there are few data of novel agents for the treatment in the text. Therefore, it give us only few new insight of the treatment for SMM. It must have more discussion and results of novel agents on SMM for the interest of readers.

3. On the prevetion therapy section, the authors could have a table to sumerized the results for the readers' convience. 

Author Response

Thank you for your thoughtful review and comments. Please see below response:

Reviewer 2:

The authors reviewed and discussed the treatment strategy for smoldering multiple myeloma, especially focused on the novel agents in recent years. The manuscript is well prepared and have referenced updated data. However, there is still some issues to be solved before publication.

  1. Though the authors have reviewed and referenced updated studies, the manuscript is too short and not qualified as a full-review articles in this journal. The authors should expand the text for more information.
  • Section 3.1 has been expanded to include discussion on immune profiling, immune microenvironment, and gene expression profiles
  • Section 4.2 text has been expanded to include updated results to date from other early & on-going trials
  • Section 6 text has been expanded to include a discussion on high-risk cytogenetic features
  • Table 3 (summary of SMM trials as prevention approach) has been added
  • Table 4 (summary results of SMM trials as treatment approach) has been added
  • Table 5 (on-going clinical trials investigating early intervention in SMM) has been added
  1. The authors aimed for the new treatment strategy in the era of novel agents, however, in addition to lenalidomide, there are few data of novel agents for the treatment in the text. Therefore, it give us only few new insight of the treatment for SMM. It must have more discussion and results of novel agents on SMM for the interest of readers.
  • Addressed- Discussion in the text on data of novel agents for treatment (section 4.2) and Table 4 (summary results of SMM trials in as treatment approach) have been added, including updated results to date from phase 1/2 trials
  1. On the prevetion therapy section, the authors could have a table to sumerized the results for the readers' convience.
  • Addressed- Table 3 (summary of SMM trials as prevention approach)

Reviewer 3 Report

Kim et al. reviewed risk stratification and clinical trials in smoldering multiple myeloma (SMM) and discussed early treatment intervention for SMM patients. The manuscript is well written, and the proposal by the authors for unanswered questions is interesting and useful for physicians, although the following points should be addressed.

Minor comments:

This manuscript is relatively short with few references for a review in Cancers.

  1. The authors mentioned an immunotherapeutic approach to the prevention of disease progression. Further, the immunomodulatory drug lenalidomide and low-dose dexamethasone could significantly prolong time to progression as well as overall survival. The authors therefore should mention the immune status of SMM patients and discuss which anti-myeloma agent would be reasonable to decrease SMM cells in specific conditions.
  2. The authors referred to the management as well as the treatment of SMM patients. The management of SMM patients in the high-risk category, especially of those with adverse chromosomal abnormalities such as del(17p), t(4;14), and t(14;16), should also be discussed.
  3. The risk factors for progression were noted. However, it would be preferable to give a more detailed discussion of potential progression-predictive markers, i.e., MM mesenchymal stem cell gene expression signatures, immune checkpoint molecules on myeloma cells, serum soluble factors, etc.
  4. Many clinical trials are ongoing or were completed in SMM patients. The authors therefore should mention the status and results of other clinical trials.

Author Response

Thank you for your thoughtful review and comments. Please see the response below:

Reviewer 3:

Kim et al. reviewed risk stratification and clinical trials in smoldering multiple myeloma (SMM) and discussed early treatment intervention for SMM patients. The manuscript is well written, and the proposal by the authors for unanswered questions is interesting and useful for physicians, although the following points should be addressed.

Minor comments:

This manuscript is relatively short with few references for a review in Cancers.

1.The authors mentioned an immunotherapeutic approach to the prevention of disease progression. Further, the immunomodulatory drug lenalidomide and low-dose dexamethasone could significantly prolong time to progression as well as overall survival. The authors therefore should mention the immune status of SMM patients and discuss which anti-myeloma agent would be reasonable to decrease SMM cells in specific conditions.

  • Addressed- expanded discussion in Section 4.2 addressing the role of immune status in SMM and studies that examined treatment approaches that are steered to directly engage the immune system to prevent progression from SMM to active disease

2.The authors referred to the management as well as the treatment of SMM patients. The management of SMM patients in the high-risk category, especially of those with adverse chromosomal abnormalities such as del(17p), t(4;14), and t(14;16), should also be discussed.

  • Addressed- Included a discussion in Section 6. Suggested approaches to the management of SMM

3.The risk factors for progression were noted. However, it would be preferable to give a more detailed discussion of potential progression-predictive markers, i.e., MM mesenchymal stem cell gene expression signatures, immune checkpoint molecules on myeloma cells, serum soluble factors, etc.

  • Addressed- Included a discussion on immune profiling, immune microenvironment, and gene expression profiles in Section 3.1 Risk factors for progression.

4.Many clinical trials are ongoing or were completed in SMM patients. The authors therefore should mention the status and results of other clinical trials.

  • Addressed- Table 3 (summary of SMM trials as prevention approach), Table 4 (summary results of SMM trials in as treatment approach), & Table 5 (on-going clinical trials investigating early intervention in SMM) have been added, including updated results to date from phase 1/2 trials. Section 4.2 text has been expanded.

Round 2

Reviewer 2 Report

The authors expand their discussion and have reviewed the novel progress on the field of SMM management. Therefore, I recommend to accept it on the journal.